# Growth Hormone Receptor Antagonist Markedly Improves Gemcitabine Response in a Mouse Xenograft Model of Human Pancreatic Cancer

**DOI:** 10.3390/ijms25137438

**Published:** 2024-07-06

**Authors:** Reetobrata Basu, Prateek Kulkarni, Deborah Swegan, Silvana Duran-Ortiz, Arshad Ahmad, Lydia J. Caggiano, Emily Davis, Christopher Walsh, Edward Brenya, Adeel Koshal, Rich Brody, Uday Sandbhor, Sebastian J. C. M. M. Neggers, John J. Kopchick

**Affiliations:** 1Edison Biotechnology Institute, Ohio University, Athens, OH 45701, USA; basu@ohio.edu (R.B.); pk585316@ohio.edu (P.K.); ds877818@ohio.edu (D.S.); duranor@ohio.edu (S.D.-O.); aa446721@ohio.edu (A.A.); lc874519@ohio.edu (L.J.C.); ed218514@ohio.edu (E.D.); cw763712@ohio.edu (C.W.); eb843318@ohio.edu (E.B.); 2Diabetes Institute, Ohio University, Athens, OH 45701, USA; 3Department of Biomedical Sciences, Heritage College of Osteopathic Medicine, Ohio University, Athens, OH 45701, USA; 4Molecular and Cellular Biology Program, Ohio University, Athens, OH 45701, USA; 5Department of Biological Sciences, Ohio University, Athens, OH 45701, USA; 6Translational Biomedical Sciences Program, Ohio University, Athens, OH 45701, USA; 7Honors Tutorial College, Ohio University, Athens, OH 45701, USA; 8Department of Cell and Developmental Biology, Vanderbilt University, Nashville, TN 37232, USA; adeel.b.koshal@vanderbilt.edu; 9InfinixBio LLC, Columbus, OH 43212, USA; rbrody@infinixbio.com (R.B.); usandbhor@infinixbio.com (U.S.); 10Department of Medicine, Endocrinology, Erasmus Medical Centre, 3015 GD Rotterdam, The Netherlands; s.neggers@erasmusmc.nl

**Keywords:** pancreatic cancer, pancreatic ductal adenocarcinoma, gemcitabine, growth hormone (GH), GH receptor (GHR), GHR antagonist, insulin-like growth factor 1 (IGF1), adjuvant, chemotherapy, chemoresistance

## Abstract

Chemotherapy treatment against pancreatic ductal adenocarcinoma (PDAC) is thwarted by tumoral activation of multiple therapy resistance pathways. The growth hormone (GH)–GH receptor (GHR) pair is a covert driver of multimodal therapy resistance in cancer and is overexpressed in PDAC tumors, yet the therapeutic potential of targeting the same has not been explored. Here, we report that GHR expression is a negative prognostic factor in patients with PDAC. Combinations of gemcitabine with different GHR antagonists (GHRAs) markedly improve therapeutic outcomes in nude mice xenografts. Employing cultured cells, mouse xenografts, and analyses of the human PDAC transcriptome, we identified that attenuation of the multidrug transporter and epithelial-to-mesenchymal transition programs in the tumors underlie the observed augmentation of chemotherapy efficacy by GHRAs. Moreover, in human PDAC patients, GHR expression strongly correlates with a gene signature of tumor promotion and immune evasion, which corroborate with that in syngeneic tumors in wild-type vs. GH transgenic mice. Overall, we found that GH action in PDAC promoted a therapy-refractory gene signature in vivo, which can be effectively attenuated by GHR antagonism. Our results collectively present a proof of concept toward considering GHR antagonists to improve chemotherapeutic outcomes in the highly chemoresistant PDAC.

## 1. Introduction

Pancreatic adenocarcinoma (PAAD) affects > 510,000 individuals and is responsible for >467,000 deaths every year worldwide, as per the 2022 estimates of the Global Cancer Observatory (World Health Organization). In the US, PAAD accounts for >50,000 cancer-related deaths (8% of all cancers) annually, with a five-year survival rate of 12.5% [https://seer.cancer.gov/statfacts/html/pancreas.html (accessed on 2 February 2024)]. Currently, it is one of the most therapy-resistant solid cancers in humans [1]. Chemotherapy, including FOLFIRINOX, gemcitabine, and nab-paclitaxel, is a major component of the current multimodal chemotherapeutic approach for pancreatic ductal adenocarcinoma (PDAC), which constitutes > 90% of pancreatic cancer cases [2]. However, PDAC is extremely chemoresistant, leading to limited therapeutic efficacy and high relapse and mortality rates, emphasizing the need for better treatment approaches [3]. Factors that endow chemoresistance in PDAC include, but are not limited to, a desmoplastic tumor microenvironment (TME), upregulated drug metabolism and efflux, epithelial-to-mesenchymal transition (EMT)-driven phenotype plasticity, and extensive metabolic reprogramming [4,5,6,7]. Therefore, it is challenging to identify a single therapeutic target that can attenuate this multifactorial treatment resistance in cancer. Over the last decade, abrogation of growth hormone (GH) action by targeting the GH receptor (GHR) to specifically curb cancer chemoresistance has shown promising in vivo results in multiple human cancer models [8]. Moreover, the GH–GHR pair and its downstream effectors are abundantly expressed in several non-tumor cell-types in the tumor microenvironment. Current knowledge of GH action in these cells indicates several putative tumor promoting and immune suppressing effects, which can be expected to be emphasized by autocrine/paracrine GH action in the TME [9,10,11]. Therefore, we postulated that antagonizing GHR can in turn improve chemotherapeutic efficacy in highly chemorefractory PDAC.

GH is an anterior pituitary hormone with postnatal whole-body endocrine actions that regulates longitudinal growth, organ development, metabolic homeostasis, and hepatic production of >70% of circulating insulin-like growth factor 1 (IGF1), another well-studied tumor-promoting factor. GH also exerts distinct autocrine/paracrine actions via multiple non-pituitary sources of origin [12,13], including the TME [14]. Following adulthood, the pituitary GH level decreases steadily with age, while non-pituitary GH production often increases [13] and drives tumor development in the colon [12,15] and breast [10,11,16]. Systemically, GH action drives insulin resistance [9] and, again, IGF1 production, both of which are detrimental factors in cancer prognosis [17]. Recently, we and others have described the covert role of GH in driving cancer chemotherapy resistance (reviewed in [8,10]) in addition to its direct tumor-promoting effects. The intracellular signaling cascade (viz. JAK2-STAT5, SRC-MAPK, and PI3K-AKT-mTOR) activated following GH/GHR binding in the TME signals induction of the EMT program, senescence, cancer stem cells, upregulation of multidrug efflux transporters, and downregulation of apoptosis [8]. Moreover, human GH is also an activating ligand of prolactin receptor (PRLR), while GH-induced hepatic and tissue IGF1 activates IGF1 receptors (IGF1Rs) and insulin receptors (IRs) abundant in the TME. Both occurrences exert overlapping and multiple tumor-promoting effects [10]. We and others have recently shown that in cancers of the liver, breast, endometrium, and melanoma, GHR attenuation markedly improves chemotherapeutic efficacy in preclinical models [10,18,19,20,21]. GH and GHR are expressed in mammalian pancreatic tissue in endothelial, epithelial, immune, and acinar cells (Appendix A), as well as in pancreatic tumor samples (Appendix A). Therefore, it is provocative to verify whether GHR antagonism as an adjuvant can improve chemotherapeutic efficacy in PDAC. Here, we report that increased GHR expression has a significant negative correlation with disease-free survival in both male and female patients with PDAC. Thereafter, we performed limited but focused in vitro and in vivo assessments of the extent of GH action in PDAC in the context of tumor chemoresistance to identify some of the underlying factors. We also identified the differentially expressed genes and related pathways in patients with high or low tumoral GHR expression (above or below mean GHR expression) in available PDAC patient databases. The findings were then corroborated at the RNA level in tumors grown in a syngeneic GH transgenic mouse model. Collectively, our results indicate that GH action indeed promotes “broad-spectrum” therapy resistance in PDAC. GHR antagonists (GHRAs) can effectively reduce tumor growth, but more importantly, improve treatment efficacy of anti-cancer compounds in both male and female mouse xenograft models. Thus, GHRA in combination with anti-cancer chemotherapy posits a promising application to treat PDAC in human patients.

## 2. Results

### 2.1. GHR Expression in PDAC and Correlation with Patient Survival

GHR is expressed in several different types of cells in the pancreas. Cell type-specific expression varies from 11% (of ductal cells) to 92% (of beta cells) in the mouse pancreas (*Tabula muris* [22], Appendix A). The expression patterns of deactivators and markers of GH signaling, CISH, and SOCS2, closely follow that of GHR expression (Appendix A). PDAC originates as transformed pancreatic acinar and ductal cells, where the GHR expression level is lowest in the normal pancreas. However, in human PDAC patients (179 samples, TCGA database), mean tumor GHR expression is 4-fold higher than that in untransformed ductal and acinar cells and comparable to the mean GHR expression of the normal whole pancreas (171 samples, TCGA and GTEx databases, Appendix A). This indicates a possible increase/overexpression of GHR in transformed compared to non-transformed pancreatic acinar and ductal cell types. Tumoral GHR expression remains stable throughout stages I to IV in the human PDAC database (Appendix A). Disease-free survival (DFS) is a reliable surrogate for overall survival in pancreatic cancer [23]. Querying one of the largest pancreatic cancer patient transcriptome profiles, which includes 1640 samples combining 16 available GEO datasets [24], we observed that increased GHR expression significantly and negatively correlates with DFS [hazard ratio (HR) = 2.16, *p* = 0.0003] and survival (9 vs. 17 months) in human PDAC patients (Figure 1A and Appendix A). Herein, the GHR-related hazard ratio is markedly higher in male (HR = 3.16, *p* = 0.0004, 10 vs. 19 months, Appendix A) than in female (HR = 1.66, *p* = 0.07, 9 vs. 15 months, Appendix A) patients. GH is also a potent ligand and activator of PRLR [21], and several subsets of patients with PDAC display upregulated PRLR expression [25,26]. In the PDAC dataset, PRLR expression also significantly correlates with increased risk (HR = 1.66, *p* = 0.0179) and poorer survival (10 vs. 15 months, Figure 1B), preferentially in male (HR = 2.8, *p* = 0.003, 10 vs. 24 months, Appendix A) than in female (HR = 0.77, *p* = 0.320) patients. Interestingly, tumoral IGF1R expression correlates with significantly increased HR and poorer overall survival, also preferentially in male patients with PDAC (HR = 1.55, log-rank *p* = 0.012, Appendix A). Additionally, we observed that in the healthy human pancreas, RNA expression of *GHR* and *IGF1* is positively correlated (Spearman correlation coefficient R = 0.43, *p* < 0.001) and the correlation further increases in pancreatic tumors (R = 0.71, *p* < 0.001) (Figure 1C). These results indicate probable GH-regulated IGF1 production in the PDAC TME. Moreover, despite no correlation in the normal pancreas (R = 0.07, *p* = 0.39) between *GHR* and *IGF2* (a potent activator of IGF1R), a much higher positive correlation in pancreatic tumors is observed (R = 0.37, *p* < 0.001) for the same (Figure 1C).

### 2.2. GH Drives Pancreatic Cancer Cell Growth

In all of our cultured pancreatic cancer cell lines (human—PANC1 and BxPC3, mouse—LTPA and Pan02), GH and GHR protein expression is confirmed by western blot (Figure 1D), while GHR protein expression is confirmed by immunocytochemistry as well (Figure 1E), indicating an autocrine loop. RNA expression of GH, GHR, IGF1, IGF1R, and PRLR is also detected using qPCR (Figure 1D), and IGF1 production in tumor cells is markedly lower in mouse tumor cells but not in human tumor cells. Moreover, in the cultured cells, the phosphorylation states of specific downstream mediators of GH signaling (STAT5, STAT3, and SRC) are upregulated following GH treatment, whereas GHRAs (either the FDA-approved GHRA, pegvisomant [27], or a novel candidate GHR antagonist, compound G [28]) suppress this effect significantly (compound G > pegvisomant) (Figure 1F and Appendix A). Interestingly, GHRA treatment in the cell lines exhibits a modest upregulation of the p38-MAPK pathway (Figure 1F and Appendix A). In this regard, activation of the p38-MAPK pathway enables gemcitabine-induced cytotoxicity in pancreatic cancer [29].

We found that exogenous GH treatment significantly increases the proliferation rate in cultured cells, which is blocked by GHRA treatment in vitro (Figure 1G). In vivo, human PANC1 cell xenografts in male nude mice show a dose-dependent increase in growth rate and final tumor weight with daily intraperitoneal (i.p.) injections of human GH compared to saline-treated controls (Figure 1H). Monotherapy with pegvisomant or compound G effectively reduces the PANC1 xenograft tumor growth rates in male and female mice (Figure 1I,J). Additionally, when 100,000 mouse Pan02 cells are implanted in syngeneic (C57BL6) bovine GH transgenic (bGH) mice with supraphysiological levels of GH and IGF1, vs. wild-type (Wt) mice, tumors develop in 100% (4/4) of bGH mice compared to only 50% (2/4) of WT mice (Appendix A). The bGH tumors are >5-fold higher in volume and weight (Figure 1K) than their WT counterparts. In human PDAC patient tumor samples, *GHR* RNA expression is positively and significantly correlated with that of several markers of GH downstream signaling, including *IGF1*, *STAT5B*, *SOCS2*, and *JAK2* (Pearson’s correlation coefficients = 0.84, 0.74, 0.73, and 0.49, respectively, false discovery rate (FDR) < 0.05; Appendix A). Gene set enrichment analysis (GSEA) of genes correlating significantly (FDR < 0.05) with GHR also show positive enrichment of multiple members of JAK/STAT signaling (Appendix A, KEGG pathway). Collectively, the data show a net definite tumor-promoting action of GH in cultured pancreatic cancer cells, xenograft samples, and human patients.

### 2.3. GHR Antagonism Markedly Improves Chemotherapeutic Efficacy against PDAC

It is currently untested whether GHRA can be administered as an adjuvant with chemotherapy in any human cancer model. We report that in cultured human and mouse pancreatic cancer cells, exogenous GH treatment partially rescues tumor cells from the cytotoxic effects of chemotherapies—gemcitabine, doxorubicin, and the EGFR-inhibitor, erlotinib (Appendix A). Both GHRAs, pegvisomant and compound G, attenuate these effects of GH and markedly restore the cytotoxic effects of the chemotherapies (Appendix A). For human PANC1 cells xenografted in male nude mice, both high and low physiologic doses of gemcitabine monotherapy markedly reduce the xenograft growth rate. But over the course of treatment, the chemotherapy effect reduces to either a cytostatic (at high dose) or no effect (at low dose) (Figure 2A). In low-dose gemcitabine monotherapy, an increase in tumor volume from day 40 (23 days post-treatment) indicates a possible onset of chemoresistance (Figure 2A). Remarkably, either pegvisomant or compound G combinations restore gemcitabine cytotoxicity, observable from day 28 (11 days post-treatment). The combination of low-dose gemcitabine and either of the GHR antagonists is significantly more efficacious than both low- and high-dose gemcitabine monotherapy (Figure 2A). The maximum therapeutic efficacy is observed in the combination regimen of compound G and high-dose gemcitabine, where 2/6 male mice in the group are tumor-free by day 43 (26 days post-treatment). In the compound G and low-dose gemcitabine combination, 1 of 6 mice is tumor free, while no mice in the high or low gemcitabine monotherapy treatment group achieve tumor clearance. Similar results are obtained in female nude mice (Figure 2B; *tumor growth is slower than in male mice*), where either dose of gemcitabine markedly lowers the tumor growth rate but fails to reverse the increase in tumor volume or achieve tumor clearance. In low-dose gemcitabine treatment, the tumor volume starts increasing by day 35 (18 days post-treatment), indicating a possible onset of chemoresistance. The co-administration of compound G (*pegvisomant combination not tested in female mice*) restores tumor sensitivity to gemcitabine. The compound G and low-dose gemcitabine combination is comparable to high-dose gemcitabine monotherapy in controlling tumor volume increase, whereas in the compound G and high-dose gemcitabine combination group, 3 of 7 mice are tumor free by day 44 (27 days post-treatment) (Figure 2B). The post-mortem final tumor weights corroborate the tumor volume measurements (Figure 2C,D). No significant differences in body weight, body composition, glucose levels, or serum IGF1 levels are observed between the nude mice groups (Appendix A). Xenograft tumors in female mice grow significantly smaller than in male mice, which could be a function of lower IGF1 levels (Appendix A) as well as the GH-antagonizing effects of estrogen in the former, but this was not confirmed here. In xenograft tumors, querying GH downstream intracellular signaling intermediates predicted from PDAC patient tumor analysis (Appendix A) show that GHRA-treated groups have significant and consistent reductions in the phosphorylation states of STAT5, STAT3, as well as SRC and AKT in both male (Figure 2E,F) and female mice (Figure 2G). ERK1/2 activation is not significantly affected in the tumors (Figure 2E,F). Interestingly, a distinct and significant increase in phosphorylation of p38-MAPK with GHRA treatment is also observed in male but not in female nude mice (Figure 2E–G). In cultured human and mouse cells, both GHRAs exhibit similar concomitant suppression of STAT5 and an increase in p38-MAPK phosphorylation states, which is more pronounced in the presence of gemcitabine (Appendix A). Notably, in pancreatic cancer, gemcitabine efficacy necessitates p38-MAPK activation [29], while gemcitabine resistance is augmented by the activation of STAT5 [30], SRC [31,32], and STAT3 [33]. Altogether, our data validate our hypothesis that GHRA combination improves chemotherapy efficacy in part by attenuating the signaling pathways responsible for gemcitabine resistance in vivo.

### 2.4. Combination of Chemotherapy and GHR Antagonism Suppresses ABC Multidrug Transporter Expression and Drug Efflux Activity in PDAC Cells and Xenografts

ATP-binding cassettes containing multidrug transporters (ABC transporters) support chemoresistance in cancer by active efflux of numerous antineoplastic agents. In pancreatic cancer, ABCG2 [34], ABCC1, ABCC5 [35], and ABCA8 [36] have been specifically reported to be associated with gemcitabine resistance, whereas additional ABC transporters are widely expressed in chemoresistant PDAC cells [37]. Given that GH action drives the overexpression of multiple ABC transporters in melanoma, liver cancer, and breast cancer [10], and our observed GHR-associated enrichment of the ABC transporter pathway in PDAC patient samples (Appendix A), here we tested and observed a strong positive correlation between RNA expression of GHR and 15 ABC multidrug transporters in PDAC patients (FDR < 0.05, TCGA) (Figure 3A). Well-known mediators of gemcitabine import (SLC28A1, SLC28A3, SLC29A1, and SLC29A2), activation (DCK, CMPK1, and NME4), or deactivation (DCTD, CDA, and NT5C) are not consistently correlated with GHR expression in the TCGA dataset (Appendix A), and also the status of gemcitabine therapy in these patients is not considered. At the protein level, in cultured cells, although exogenous GH treatment does not have a consistent effect on ABC transporters, each of the GHRAs robustly suppress the expression of ABCA8, ABCB1, ABCC1, ABCC2, and ABCG2 (Figure 3B and Appendix A), further supporting the presence of an autocrine/paracrine GH action (Figure 1D) and the ability of GHRAs to abrogate the same. Also, gemcitabine treatment itself strongly induces the expression of multiple ABC transporter proteins (ABCA8, ABCB1, ABCG2), and a distinct attenuation of this effect is observed in the presence of GHRAs (Appendix A). Moreover, pancreatic cancer cells in culture show a >2-fold increase in the rate of drug efflux and a reduction in the amount of intracellular drug surrogate and ABC transporter substrate [DiOC(2)3] retention following GH treatment, whereas both GHRAs completely suppress the same and show increased retention of the fluorescent compound (Figure 3C and Appendix A). Several of the top positively correlated genes from Figure 3A are tested with RT-qPCR and are found to be differentially upregulated by GH and chemotherapy treatment in xenograft tumors from nude mice (male and female) and also in male bGH vs. WT mice (Figure 3D and Appendix A). Interestingly, GHRA monotherapy increases *GH1* and *PRLR* expression most significantly in the tumors of male mice (Appendix A), whereas in the presence of low-dose gemcitabine, *GH1*, *GHR*, *IGF1*, *IGF1R*, and *PRLR* expression is increased in both male and female mice (Appendix A). Monotherapy with either GHRA reduces RNA expression of *ABCA6*, *ABCA8*, *ABCA9*, *ABCB1*, *ABCG2*, *ABCC1*, *ABCC2*, *ABCC4*, *ABCC5*, and *ABCC9* in nude mice, wherein compound G shows a higher suppressive effect than pegvisomant (Figure 3D). Additionally, gemcitabine induces the upregulation of ABC transporter RNA expression, but not in the presence of GHRA. During combination treatment with gemcitabine, compound G is effective in suppressing the increase in chemotherapy-induced ABC transporter expression in both sexes (Figure 3D, *pegvisomant is tested only in male and not in female mice*). Moreover, as postulated, the RNA levels of most ABC transporters assessed are increased in the tumors of bGH mice compared to those of WT mice, showing that, even in the absence of chemotherapy, tumors exposed to high GH and IGF1 levels exhibit intrinsic chemoresistance (Figure 3D). In male and female nude mice xenografts treated with GHRAs, western blots further confirm a marked and consistent suppression of ABC transporter proteins (ABCA6, ABCA8, ABCB1, ABCC1, and ABCG2) implicated in gemcitabine resistance in human patients and with a strong positive correlation with tumoral GHR expression (Figure 3E–G). Protein levels of ABCB1, ABCA8, and ABCG2 are also significantly upregulated in bGH mouse tumors compared to those in WT mice (Figure 3H). Collectively, our data show that GH action upregulates chemoresistance in PDAC as a function of increased ABC transporter expression and drug efflux, and GHRA treatment can reverse this outcome and appears to be one of the factors underlying our in vivo observation of restoring chemosensitivity in PDAC.

### 2.5. Combination of Chemotherapy and GHR Antagonism Suppresses EMT Induction in PDAC Cells and Xenografts

Commitment to the EMT program is central to gemcitabine chemoresistance in pancreatic cancer [38,39]. GH is a potent driver of EMT in normal and transformed cells [40] and has also been shown to promote EMT in pancreatic cancer cells in vitro [41]. In PDAC patient transcriptomes, we observe a strong correlation between GHR expression and that of multiple EMT transcription factors and markers, including *ZEB1*, *ZEB2*, *Snail* (*SNAI1*), and activating components of the EMT-inducing TGFb signaling pathway (*TGFB3*, *TGFBR1*, *TGFBR2*, and *TGFBR3*) (Figure 4A). In cultured cells, GH consistently upregulates the protein expression of EMT transcription factors *ZEB1* and Snail (*SNAI1*) (Figure 4B and Appendix A). However, in all cases, GHRA treatment suppresses EMT markers to below baseline levels, indicating the suppression of the autocrine/paracrine GH action (Figure 4B and Appendix A). Accordingly, GH increases the basement membrane invasion capacity, which is suppressed by both pegvisomant and compound G (Figure 4C). The number of colony-forming units following either doxorubicin or gemcitabine treatment is also upregulated by GH and suppressed by pegvisomant (Figure 4D). Gemcitabine treatment alone strongly induces the expression of EMT transcription factor protein expression, but in presence of GHRAs, this effect is attenuated (Appendix A). Next, in xenograft tumors, we assess the RNA expression of EMT mediators that are positively correlated with GHR expression in human tumor samples (Figure 4A). In bGH mouse tumors, multiple EMT factors, similar to that observed in human patients (in Figure 4A), are upregulated by >2-fold compared to their WT counterparts even in the absence of chemotherapy (Figure 4E). In the case of GHRA monotherapy in the nude mouse model, compound G is more effective than pegvisomant in suppressing the increase in mesenchymal marker RNAs for *CDH2*, *SNAI2*, *TGFB3*, *TWIST* (*1* and *2*), and *ZEB1* in male mice (Figure 4E). Gemcitabine treatment induces a strong increase in all EMT markers tested, which is attenuated by the presence of GHRAs in both sexes (Figure 4E). Furthermore, both GHRAs show consistent suppression of the protein (Figure 4F–H) expression of mesenchymal markers ZEB1, SNAI1, CDH2, and VIM and consistent upregulation of the epithelial marker CDH1 in vivo. On the other hand, bGH mouse tumors show increased expression of EMT transcription factors (ZEB1, SNAI1, and SNAI2) and mesenchymal markers (CDH2 and VIM) compared to WT tumors (Figure 4I). Our in vitro and in vivo analyses validate that targeting GH action effectively attenuates EMT program activation, especially under chemotherapeutic challenge, in pancreatic tumors and supports our hypothesis that GHRA promotes chemo-sensitization in PDAC.

### 2.6. The Combination of Chemotherapy and GHR Antagonism Suppresses Markers of Apoptosis, Fibrosis, Lymphangiogenesis, Angiogenesis, Senescence and Stemness in PDAC Cells and Xenografts

PDAC tumors are poorly responsive to several therapeutic modalities owing to a desmoplastic TME, characterized by extensive extracellular matrix (ECM) remodeling and fibrosis [42,43,44]. Additional factors include suppression of apoptosis [45], dysregulated vascular and lymphatic endothelial action [46,47], induction of senescence and immunosuppressive effects from the senescence-associated secretory phenotype (SASP) [48], induction of stemness [49], and drug-metabolizing enzymes, such as cytochrome P450s [50]. Based on the well-studied effects of GH in enhancing several of these pathways across multiple cancer types [10], we first compared the effect of GH on the expression of corresponding gene modules in the human PDAC patient database (TCGA). Gene sets, representing the established markers of each molecular mechanism, are arranged into individual modules, viz. *Fibrosis*, *Apoptosis*, *Senescence*, *Angiogenesis*, *Lymphangiogenesis*, *Stemness*, and *CYP450s*. Tumors with high GHR (>mean) expression display a strong positive correlation with several marker genes of each of these pathways (Figure 5). ECM remodeling and fibrosis-related genes underlying the desmoplastic TME of PDAC are most strongly and positively correlated with GHR expression, indicating that GH may be an important mediator of desmoplasia in the pancreatic TME. Cancer-associated fibrosis promotes therapeutic resistance and immune evasion in pancreatic cancer [43,44,51,52] and recent evidence implicates GH as a covert driver of multi-tissue fibrosis [8]. Here, firstly we observe a high positive correlation between RNA expression of *GHR* and that of *FAP*, *ACTA2*, and multiple collagen (*COL*), matrix metalloproteinase (*MMP*s), and tissue inhibitor of metalloproteases (*TIMP*s) genes (Figure 5A). Secondly, higher *GHR* expression correlates positively with anti-apoptotic *BCL2* and negatively with several pro-apoptotic mediators (*BIK*, *BAD*, *BID*, *BAX*, *BAK1*, *CASP6*, and *CASP7*) in the patient samples (Figure 5B). GHR expression in lymphatic endothelial cells is many times higher than that in blood vascular endothelial cells, and GH is known to promote lymphangiogenesis in vitro and in vivo [53], while forced autocrine GH expression increases vascular and lymphatic micro vessel density in human breast cancer xenografts [11]. In the human PDAC patient tumor transcriptome (TCGA), we identify strong and significant correlations between tumoral *GHR* expression and known lymphangiogenic (*LYVE1*, *FLT4*, *VEGFC*, and *PDPN*) [47] and angiogenic (*PDGFRa*, *FGFR1*, *TGFB3*, *TGFB1*, *TGFBR2*, *HIF1a*, *IL6*, and *IL1B*) [54] markers (Figure 5C,D). Furthermore, chemotherapy accelerates the development of sub-populations of senescent cells in the TME, which promotes widespread immune suppression via the secretion of a cocktail of immunoinhibitory cytokines, known collectively as SASP [48]. In human PDAC patient tumors, we observe that several SASP factors, such as *GH1* [55], *IL10*, *IL1b*, *IL6*, *CCL2*, and *CCL5*, which exert primary immune suppression (effector T cell inactivation and regulatory T cell production), are significantly and positively correlated with GHR expression (Figure 5E). Moreover, markers of cancer stem cells, *ALDH1* and *CXCR4*, are strongly associated with *GHR* in PDAC (Figure 5F). We also observed that a number of cytochrome P450s (known targets of hepatic GH action and drug modulators) are upregulated in PDAC patient tumors, with *CYP1B1*, *CYP2U1*, and *CYP7B1* displaying the highest correlations and *CYP2W1* having the lowest correlation with GHR expression (Figure 5G). Incidentally, CYP2W1 is important for the activation and response to chemotherapy, such as mitotane in cancer [56], whereas high expression of CYP1B1 is known to drive gemcitabine resistance in pancreatic cancer [50]. We performed a limited verification of these findings from the transcriptome analysis of human PDAC patients, using RNA from Pan02 murine tumor allografts in bGH and WT mice and performed qPCR for selected genes. Despite species differences, treatment naivety, and differences in tumor genetics, the mouse tumors exposed to supraphysiological levels of GH and IGF1 (as in bGH mice) show increased expression of markers of ECM remodeling and fibrosis (*Timp2*, *Timp3*, *Timp4*, and *Col6a3*), anti-apoptosis (*Bcl2*), lymphangiogenesis (*Flt4*, *Lyve1*, and *Pdpn*), angiogenesis (*Vegfa*, *Il1b*, *Pdgfra*, *Tgfb3*, and *Tgfb1*), senescence (*Il13*, *Il6*, *Pdl1*/*Cd274*, *Ccl5*, *Il10*, *Il1b*, and *Ccl2*), stemness (*Thy1*), and CYP450s (*Cyp2u1* and *Cyp1b1*) (Figure 5H and Appendix A), confirming the immune-suppressive and therapy-resistant actions of GH in PDAC. Moreover, tumoral *Pdl1* (*Cd274*) but not *Pdl2* (*Pdcd1lg2*) expression is elevated by almost 6-fold in bGH tumors (Figure 5H and Appendix A). In marked corroboration, we also found that in 154 tumor samples from patients with PDAC (TCGA cohort): (i) gene set enrichment analysis (GSEA) of GHR-correlated genes show enrichment of the ‘cancer immunotherapy by PD-1 blockade’ pathway (Appendix A); (ii) out 30 cancer types, GHR is strongly and positively correlated with the expression of 17 out of 23 immune-inhibitory cytokines in PDAC (Appendix A); (iii) high tumor infiltration of immune-suppressive lymphocytes (T-reg, MDSCs, and macrophages) in GHR-enriched PDAC tumors (Appendix A); and (iv) enrichment of several (13 out of 23 top enriched pathways) immune-related pathways correlate with tumoral GHR expression (Appendix A). Together, our data indicate that in addition to direct chemorefractory effects on the tumor, GH action may promote desmoplasia and host immune evasion in PDAC.

## 3. Discussion

Targeting GHR in multiple types of human cancers has been hypothesized and shown to significantly attenuate tumor resistance to multiple antineoplastic approaches, including radiation therapy, chemotherapy, targeted therapy, and immunotherapy [10]. GHR is overexpressed in human PDAC. The chemotherapy-sensitizing effects of direct GHR antagonism in preclinical models of human melanoma, breast cancer, and liver cancer exhibit a consistent suppression of ABC multidrug transporters and markers of EMT, leading us to hypothesize that targeting GHR in PDAC can reverse the chemoresistant phenotype of this cancer. To date, one isolated study has reported the anti-cancer properties of RNAi-mediated GHR silencing in cultured human pancreatic cancer cells [41], and no studies have reported the in vivo use of GHR antagonism or a combination of GHRAs with commonly used chemotherapies to improve therapeutic outcomes in any cancer. Here, we report the first preclinical validation of the postulated gemcitabine-sensitizing effect of GHR antagonism using two different GHRAs in cultured pancreatic cancer cells and mouse tumor xenografts. GHR antagonism rescued gemcitabine resistance and improved gemcitabine efficacy in vivo, wherein the GHRA-treated groups had marked tumor retraction in multiple mice of both sexes. Drug–drug interaction (DDI) measured by the Chou–Talalay method (PMID: 20068163) showed that coefficients of DDI at the end of the study were 0.875, 0.787, and 0.597 for pegvisomant + 20 mg/kg gemcitabine, compound G + 20 mg/kg gemcitabine, and compound G + 80 mg/kg gemcitabine, respectively, in male mice. By contrast, female mice showed only an additive effect and no synergy for the dose of compound G used. Our analyses of the human PDAC tumor transcriptome showed that increasing GHR expression correlates positively with several gene expression modules, which collectively confer therapy resistance and immunosuppression in patients. Additionally, we compared these results in a syngeneic mouse model of excess versus normal GH expression, and the majority of observations of therapy-refractory gene expression changes were corroborated. Overall, our results indicate GHR antagonists in combination with gemcitabine is a promising modality to significantly potentiate the efficacy of the existing anti-PDAC regimen. This is particularly important, as gemcitabine is one of the most widely used chemotherapies in PDAC [57] and often displays a blunted efficacy in patients [58].

Relevance of targeting GHR in pancreatic cancer: GHR is expressed in all pancreatic cell types, albeit variably, with the highest level (92% of cells, mean expression = 4.4) in pancreatic beta cells and the lowest level (11% of cells, mean expression = 0.5) in pancreatic ductal cells of the normal pancreas (Appendix A). PDAC originates mainly from transformed ductal cells of the pancreas and displays GHR expression comparable to that of the whole pancreas (all cell types combined). This indicates that the GHR expression level is supraphysiological in transformed ductal cells compared with that in normal ductal cells of the pancreas. Notably, *IGF1R* RNA expression, in addition to having a significant inverse correlation with human male PDAC patient survival, closely follows *GHR* expression in the mouse pancreatic cell types (Appendix A) and the human PDAC transcriptome, indicating active local IGF1R signaling in the pancreas, which is detrimental to the TME [59]. Additionally, gemcitabine dose-dependent activation of IGF1R drives gemcitabine resistance in pancreatic cancer cells by inducing EMT-mediated phenotype switching [60]. Both IGF1 and IGF2 are potent activating ligands for IGF1R. Of relevance, a Pfizer clinical trial showed that 40 mg daily pegvisomant treatment for 14 days caused sustained reductions in both serum IGF1 and IGF2 by 33% and 35%, respectively, in healthy human subjects [61]. Therefore, appropriate doses of GHRAs that result in IGF1 lowering can be expected to significantly attenuate the tumor-promoting effects mediated by IGF1R activation. IGF1R attenuation is achievable by IGF1R inhibitors, and IGF1 is deemed a provocative target in anti-cancer research, supported by a plethora of studies [62]. However, IGF1R targeting in cancer [63] has proven to be a consistently ineffective strategy in controlling tumor progression, as shown in 183 human clinical trials involving >12,000 patients across several tumor types (2003–2021) at a net cost estimated between $1.6–2.3 billion [64]. A major underlying factor behind these results is that IGF1R inhibition increases GH production due to de-repression of the IGF1-mediated negative feedback loop in the pituitary, as well as potentially allowing free IGF1 and IGF2 to activate the insulin receptor (IR) in the tumors. Therefore, it is provocative to test the use of GHRAs as a case-specific maintenance protocol against tumor relapse or as a preventive regimen in high-risk cases of age-associated tumor development promoted by local GH action in tissues [13] (not tested here).

It is important to note that the GHRA pegvisomant does show physiological effects in vivo, independent of IGF1 reduction, even at doses five times lower than the dose used here [65]. The GHRA dose in the current study is thus physiologically relevant but lower than most previously used doses of pegvisomant in nude mouse tumor xenograft studies aimed at causing significant serum IGF1 reduction in mice. Our current GHRA dose was chosen to not cause any marked suppression of serum IGF1 in the mice, i.e., it did not block the endocrine mouse GH action. Therefore, our results preferentially reflect the effects of blocking IGF1-independent and autocrine/paracrine actions of GH in the tumor milieu. Future studies with higher dosages of the long-acting pegylated GHRAs used in this study can, in addition to blocking direct GH actions on the tumor, also suppress systemic IGF1 levels. The effects of such escalated dosages on treatment outcomes need to be ascertained. Focused preclinical studies with human PDX models with dose-titration of combinations of GHRAs and selected antineoplastic agents and subsequent carefully designed human clinical trials are needed to validate the transformative prognostic potential of targeting GHR in pancreatic cancer. Recent mechanistic combinatorial targeted therapy approaches, such as double inhibition of IGF1R and ERK signaling in organoid models, display increased sensitivity of pancreatic tumors to autophagy inhibitors such as hydroxychloroquine [66] and serve as a precedence to our concept. Moreover, given the role of PRLR activation in PDAC and human GH as a potent ligand for PRLR, new dual inhibitors of GHR and PRLR [28,67] provide exciting opportunities.

Relevance of targeting GH in the tumor microenvironment: We and others have described a covert action of GH–GHR in cancer, not only in increasing cell proliferation and decreasing tumor cell apoptosis, but also in driving therapy resistance against radiation therapy, chemotherapy, targeted therapy, and immunotherapy [8,10,68]. This is the first study to extend the validity of our observations to the therapeutically challenging case of PDAC. We showed here that GH drives several therapy-resistance mechanisms in PDAC, focusing mainly on the GH-induced increase in ABC transporter expression and induction of the EMT pathway, which are only part of the molecular mechanisms underlying gemcitabine resistance in cancer. Indeed, in the last five years, multiple ABC transporters have been directly implicated in gemcitabine resistance in PDAC, of which ABCA8 and ABCG2 were consistently suppressed by GHRA. In the present study, we observed a marked suppression of gemcitabine resistance in the presence of GHRAs. We have previously shown that GHR inhibition sensitizes melanoma cells to paclitaxel, as well as platinum-based antineoplastic drugs like cisplatin [18,69,70] and the pyrimidine analog, fluorouracil (5-FU) [70]. Therefore, based on our previous and current studies, GHRAs can potentially augment current anti-PDAC chemotherapeutic combinations [71], such as gemcitabine-paclitaxel as well as FOLFIRINOX (combination of 5-FU, oxaliplatin, irinotecan, and folinic acid). In addition, several components of the TME express GH and GHR and have well-characterized GH-regulated phenotypes, which are tumor-supportive, therapy-refractory, and immune-suppressive. Future studies directed at the effects of GHRA treatment in managing these effects are critical and necessary. 

Based on the empirically demonstrated role of GH in other cancer studies, we identified that GHR antagonism in PDAC leads to suppression of the gene signature of hallmark cancer processes. Our analyses of human PDAC patient data showed that desmoplasia and fibrosis, which constitute major roadblocks in PDAC therapy, correlate strongly with GHR expression. Moreover, a positive correlation was observed between GHR and markers of cancer stem cells, apoptosis inhibition, gemcitabine-deactivating CYP450s, increased angiogenesis and lymphangiogenesis, and senescence. Transcriptome analysis of 269 PDAC patients showed that the lymphatic lineage of endothelial cells in the pancreatic TME is maximally associated with poor overall survival (HR = 2.91, *p* = 0.003) and progression-free survival (HR = 4.65, *p* < 0.001) [72]. As GH promotes lymphangiogenesis and the corresponding markers in PDAC correlate with GHR expression in our analyses, it can be expected that GHRAs can lower or inhibit tumor lymphangiogenesis to attenuate tumor growth, which remains to be tested in PDAC. Lastly, we reveal a provocative association of GH action with the cancer immunosuppressive TME. Tumoral GHR expression in patients correlated strongly and positively with SASP components, immunoinhibitory markers, immunosuppressive TILs, and immune-modifying pathways. Some of these findings were validated in the syngeneic tumor-bearing bGH mice. Our results align with recent findings of a lack of response to anti-PD1/PDL1 therapy in patients with liver and gastric cancers with high levels of circulating GH [10].

Our observations are limited by bulk RNA estimations of tumor samples and should be treated as a springboard for future validation studies with higher resolution methods. It is imperative to clarify the extent of GH signaling and the effect of GHR antagonism on individual cell types in the TME of PDAC via spatial single-cell transcriptomic and proteomic methods. Moreover, we have not studied longer term post-treatment effects (tumor relapse, survival) in our mice or the details of observed sex-specific differences—which are essential to assess prior to clinical translation. Although not studied in this report, higher serum glycine levels reduced the risk of pancreatic cancer by 70%, as observed in a prospective cohort study in human patients [73]. In this regard, bGH transgenic mice, which develop spontaneous neoplasms at a higher rate than WT mice [74], also have markedly lower glycine levels than their WT counterparts, while the reverse is observed in GHR-knockout (GHRKO) mice, which are protected from cancer [75].

In conclusion, supported by immunodeficient nude and syngeneic mouse tumor models, cell culture, and bioinformatic analyses of human pancreatic cancer patient transcriptome data, this study presents a robust and, to our knowledge, the first validation of combining GHRAs in a chemotherapy regimen in pancreatic cancer that markedly improved therapy efficacy. We emphasize that although we validated our hypothesis of GHRA-mediated chemo-sensitization in this study, GHRAs have direct effects on the non-tumor components of the TME in promoting therapy resistance and immune suppression, which are yet unknown. Moreover GHRAs are also poised to exert indirect effects by lowering the level of serum IGF1 (and IGF2), the activating ligands for insulin and IGF1 receptors, as well as improving insulin sensitivity in patients [76,77,78,79]. Rarely can an antitumor agent exert such a versatile effect. Finally, with the existence of an FDA-approved GHR antagonist and several others in active development targeting the full spectrum of GH action [28,67,80], our current study sets course in a feasible paradigm shift in the therapeutic augmentation of pancreatic cancer. 

## 4. Materials and Methods

Cell culture and treatments: Pancreatic cancer cell lines PANC1 (human male, CRL-1469), BxPC3 (human female, CRL-1687), and LTPA (mouse female, CRL-2389) were purchased from ATCC (Manassas, VA, USA), and Pan02 (mouse male) cells were obtained from Charles River (Wilmington, MA, USA). Cells were maintained in RPMI-1640 or DMEM supplemented with 10% fetal bovine serum (FBS) and 1X penicillin-streptomycin, all purchased from ATCC, in a humidified 5% CO_2_ incubator at 37 °C. All cell treatments were performed in <5 months from purchase of the cells and within passage numbers 4–12. No mycoplasma contamination was detected in the cell cultures when tested at the beginning and end of the study using the Universal Mycoplasma Detection Kit (ATCC, 30-1012K). Treatment with the indicated concentrations of recombinant human GH (Antibodies Online, Limerick, PA, USA), chemotherapy [gemcitabine, doxorubicin, etoposide, all purchased from Selleckchem (Houston, TX, USA)], pegvisomant (a gift from Dr. Sebastian Neggers), or compound G (InfinixBio, Columbus, OH, USA) were performed in 2% FBS-supplemented media for the given timepoints, as described previously [81]. In the in vitro studies, all treatments with anti-cancer drugs were performed at the EC50 values determined at the beginning of the experiment, as shown in Appendix A. All in vitro studies were performed using compound G (GGSSG-hGH-G120K-T142C-dPEGA-H151C-dPEGA), and all in vivo studies used compound G’ (M-hGH-G120K-T142C-dPEGA-H151C-dPEGA), as described previously [28]. Both are G120K substituted versions of human GH with site-specific discrete PEGylation at cysteines substituted at T142 and H151 away from the GH–GHR interaction region [28]. In this study, both compounds were referred to as compound G.

Mouse xenograft studies: Four million human PANC1 cells in 100 µL of 1:1 *v*/*v* mixture of PBS and Matrigel Growth Factor Reduced Matrix (Corning, #356231, Santa Barbara, CA, USA) were subcutaneously engrafted on the right flank of 6-month-old immunocompromised nude mice (NU/J, RRID: IMSR_JAX:002019, Jackson Laboratory, Bar Harbor, ME, USA). Following tumor growth stabilization (day 17, after the implanted tumors reached a size of 100–200 mm^3^), the animals were randomized into groups and treatments initiated with either a low (20 mg/kg BW/3 days) or a high (80 mg/kg BW/3 days) dose of gemcitabine, with or without once-daily administration of either GHRA pegvisomant or compound G at 10 mg/kg BW/day. All treatments were by intraperitoneal (i.p.) injection (each injection volume = <200 µL) alternating each day at either the right or left peritoneum. The lengths of the perpendicular tumor diameters were measured every 3 days using a digital caliper, as previously described. Tumor volume was calculated using the following formula: tumor size = 0.5 × (length) × (breadth)^2^. The mice were sacrificed when the tumor volume exceeded 1000 mm^3^ and excised tumors were either flash frozen in liquid nitrogen and stored at −80 °C for subsequent RNA and protein extraction or fixed in 10% formalin for 8 h and transferred to 70% ethanol before being processed for immunohistochemistry (IHC). For nude mice, there were 6–8 mice per group in each of the male and female mouse groups. For the syngeneic mouse study, 100,000 Pan02 cells in 100 µL of 1:1 *v/v* mixture of PBS and Matrigel Growth Factor Reduced Matrix were subcutaneously engrafted on the right flank of immunocompetent, 6-month-old bovine GH transgenic (bGH) mice and their wild-type (WT) counterparts. There were 4 mice per group in the syngeneic mouse study with bGH vs. WT mice. All animal studies were performed in compliance with the policies approved by the Ohio University Institutional Animal Care and Use Committee.

Cell viability assay: This was performed to assess the effects of chemotherapy, and/or GH, and/or GHRA treatment on cell viability in vitro, as described previously [82,83]. Experiments were performed in triplicate. Briefly, cells were plated at 10,000 cells/well in a 96-well plate. After overnight incubation, cells were treated as described for the specified amount of time. Cells were incubated in 2% FBS-supplemented media containing 2.5 nM/50 ng/mL (or otherwise mentioned) recombinant human (for human cells) or bovine (for mouse cells) GH and/or 500 nM pegvisomant or compound G for 72 h. Finally, the media was replaced with resazurin reagent (Abcam, cat#129732) and incubated for up to 2 h, and absorbance at 570 nm (reference wavelength 600 nm) was measured using a Spectramax250 (Molecular Devices, San Jose, CA, USA) spectrophotometer with SoftMaxPro v4.7.1 software. Three individual experiments were performed on different days.

Cell migration assay: This assay was performed to assess the effect of GH and/or GHR antagonists on the migration capacity of tumor cells, as described previously [82]. Experiments were performed in triplicate. Cells were seeded at 30,000 cells per well in a 24-well plate and a 200 µL pipette tip was used to streak and clear all cells along the midline of each well. Following a wash with PBS, the cells were incubated in 2% FBS-supplemented media containing 2.5 nM/50 ng/mL (or otherwise mentioned) recombinant human (for human cells) or bovine (for mouse cells) GH and/or 500 nM pegvisomant or compound G for up to 48 h and imaged every 24 h. The total uncovered area at the start and end of the assay was imaged using a BioTek Cytation-3 microplate imager (Gen5 v2.09.2 software) and quantified using ImageJ software (v1.53k, NIH) [83]. Three individual experiments were performed on different days.

Cell invasion assay: This assay was performed to assess the effect of the hGHR antagonist on hGH-induced invasion of cancer cells. Cells were pretreated with GH in DMEM containing 2% FBS-containing DMEM or EMEM for 48 h. The CytoSelect 96-well Cell Invasion Assay kit (CBA-112, Cell Biolabs, Inc., San Diego, CA, USA) was used according to the manufacturer’s instructions. Briefly, on the day of the assay, the cells were trypsinized, counted, and 100,000 cells were seeded per well in the upper chamber of the 96-well invasion assay well coated with basement membrane and incubated with the respective treatments in serum-free media for 24 h. After 24 h, the cells under the membranes were dislodged, lysed, and stained with CyQuant GR dye solution. The fluorescence intensity correlated with invasive cell number was measured at ex 480 nm/em 520 nm using a fluorescence plate reader. Three individual experiments were performed on different days.

Drug retention assay: This assay was performed to measure the effect of the hGHR antagonist on the hGH-induced drug efflux property of pancreatic cancer cells in culture. Briefly, pancreatic cancer cells were treated for 7 days with 2.5 nM hGH or 2.5 nM hGH + 500 nM hGHR antagonist in 2% FBS-containing growth medium, with the medium being refreshed every 24 h. On the day of the assay, the cells were trypsinized, counted, and suspended in cold DiOC_2_(3) dye on ice for 30 min (Millipore Sigma, Burlington, MA, USA). The cells were then centrifuged, the supernatant was removed, and cell pellets were resuspended in cold efflux buffer and distributed into different Eppendorf tubes under the following conditions: one set of tubes was kept on ice as negative controls, while the other two sets were kept in a 37 °C water bath for 30 and 120 min, respectively, allowing the active drug efflux pumps to drive out the DiOC_2_(3) dye. The cells were then washed, resuspended, and the cell suspensions were dispensed into the wells of a black-walled 96-well plate, and fluorescence was measured using a fluorescence plate reader at an excitation wavelength of 485 nm and an emission wavelength of 530 nm. Three individual experiments were performed on different days.

RNA extraction, quantification, and RT-qPCR were performed as previously described [69,82]. Briefly, after the described treatments, the total RNA from cultured cells was extracted using the IBI Total RNA Extraction Kit (IBI Scientific, Dubuque, IA, USA) or from tumor tissues using the GeneJET RNA Purification Kit (Thermo Fisher, Waltham, MA, USA) following the manufacturer’s protocols. From the extracted RNA, mRNA levels were quantified by oligo-dT reverse transcription using a High Capacity cDNA Reverse Transcription Kit (Thermo Fisher), followed by quantitative PCR using Applied Biosystems reagents following the manufacturer’s protocols. The primer sequences are provided in the Appendix A.

Protein extraction, quantification, and western blot: These procedures were performed as described previously [62,75]. Briefly, lysis of cultured cells or tumor tissues was performed using 1.5× RIPA buffer (Millipore Sigma, Burlington, MA, USA) supplemented with 1× phosphatase-protease inhibitor cocktail (Cell Signaling Technology, Danvers, MA, USA). Protein concentration in the lysates was estimated using a Bradford assay and 80 µg of protein was loaded onto 4%–16% gradient denaturing gels, transferred to 0.2 µm PVDF membranes (Thermo Fisher), blocked with freshly prepared 5% non-fat dry milk solution in 1× TBS-T buffer, and probed using target-specific primary and secondary antibodies. A list of antibodies and their sources is provided in the Appendix A.

Bioinformatic analyses: We quantified the Pearson correlations of GHR RNA expression with that of all other genes in the TCGA dataset of 154 patients with pancreatic ductal adenocarcinoma (HiSeq RNA at UNC; pipeline: Firehose_RSEM_log2) using the Linkedomics platform [84]. Pearson correlation coefficients of groups of genes that were differentially and significantly (FDR < 0.05) upregulated with increasing GHR expression are represented in the heatmap. We further performed gene set enrichment analysis (GSEA; Broad Institute, Cambridge, MA, USA) [normalized TPM values from TCGA PDAC cohort against the Molecular Signatures Database (v7.1 MSigDB)] to identify pathways and ontologies enriched in the high GHR group of patients. The parameters for enrichment analysis included: minimum number of IDs in the category: 5; maximum number of IDs in the category: 2000; significance level: top 25; and number of permutations: 1000. The KMplotter platform [24] was employed for survival analysis and the TISIDB platform [85] was used for tumor–immune interactions.

Statistical analyses: For all in vitro experiments that compared one parameter between multiple sets of samples, statistical significance was determined by unpaired two-sided Students *t*-test for two groups or by one-way or two-way ANOVA with Tukey’s multiple comparisons test for three or more groups using GraphPad Prism 6 software. Mouse sample sizes were estimated using the resource equation method. Calculating with the maximum degrees of freedom (at 20), at least 6 animals per group in the nude mice study was required. We started with 8 animals/group to ensure completion with at least 6 animals/group in order to reach significance, accounting for accidental loss of animals during treatment. According to the previous power calculation based on normal distributions and the magnitude of changes observed in preliminary studies, we calculated that a sample size of 8 in each group would have 90% power (α < 0.05) to detect a significant difference in means of the parameters to be studied. For the mouse experiments, repeated measures ANOVA combined with Bonferroni’s multiple comparison was used to compare tumor volumes across treatment groups at each measured timepoint. Western blot experiments were compared using one-way or two-way ANOVA with Tukey’s multiple comparisons test using GraphPad Prism 6 software (v7.04).

## Figures and Tables

**Figure 1 ijms-25-07438-f001:**
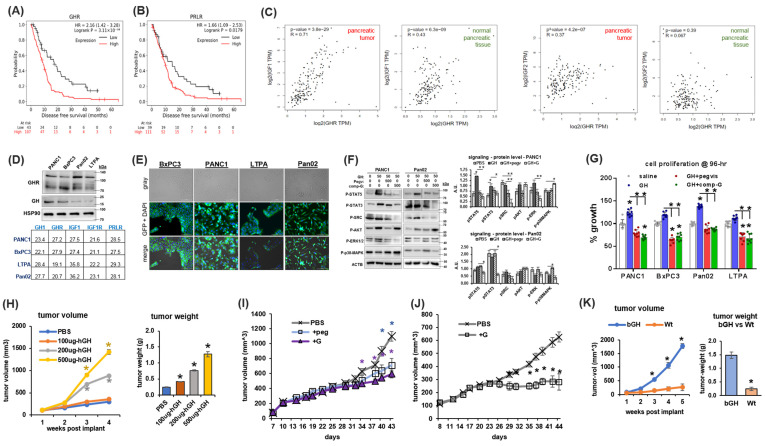
GH action is a driver of pancreatic cancer progression and inversely correlates with disease-free survival in human PDAC patients. (**A**,**B**) Kaplan–Meier survival plot depicting correlation of human GHR (**A**) or PRLR (**B**) RNA expression with disease-free survival in human pancreatic ductal adenocarcinoma patients [data re-analyzed using KMplotter from Posta and Gyorffy, *Clinical and Translational Science*, 2023, doi: 10.1111/cts.13563]. (**C**) Spearman correlation of GHR expression vs. IGF1 (*left*) and IGF2 (*right*) expression in pancreatic tumor and normal pancreatic tissue in PDAC patients (TCGA and GTEx) [data generated using GEPIA2 portal]. (**D**) (*top*) Western blot (representative blots) showing protein expression of GH and GHR in cultured pancreatic cancer cell lysates, and (*bottom*) the mean threshold cycle (Ct; *higher Ct means lower expression*) values from quantitative PCR showing relative RNA expression of GH1, GHR, IGF1, IGF1R, and PRLR in pancreatic cancer cultured cells. (**E**) Immunocytochemistry for GHR (green) in cultured pancreatic cancer cells with nuclei stained with DAPI (cells in grayscale) (magnification = 20X, scale bar = 100um). (**F**) Western blot of GH downstream intracellular signaling mediators—STAT5, STAT3, SRC, AKT, and ERK1/2—in cultured pancreatic cancer cells treated with either of saline, GH (50 ng/mL), GH (50 ng/mL) + pegvisomant (500 nM), or GH (50 ng/mL) + compound G (500 nM). (**G**) Effects of 96 h treatment with the above concentrations of GH and GHR antagonists (pegvisomant and compound G) on pancreatic cancer cell growth in culture. For the above, significant differences were compared using two-way ANOVA with Tukey’s multiple comparisons test and * signifies *p* < 0.05. (**H**) Increases in tumor volume and final tumor weight in male nude mice (n = 4) treated with 25, 50, or 125 ug/day of recombinant human GH by intraperitoneal (i.p.) injection. (**I**,**J**) Effects of treatment with GHR antagonists (pegvisomant or compound G) on PANC1 xenograft tumor volume growth in male (**I**) and female (**J**) nude mice (n = 6). (**K**) Increase in murine Pan02 allograft tumor volume with time and final tumor weight in male syngeneic C57BL6 mice—wild-type (Wt) or transgenic for bovine GH (bGH; high GH, high IGF1) (n = 4). Significant differences were compared using repeated measures ANOVA combined with Bonferroni’s multiple comparison and * signifies *p* < 0.05.

**Figure 2 ijms-25-07438-f002:**
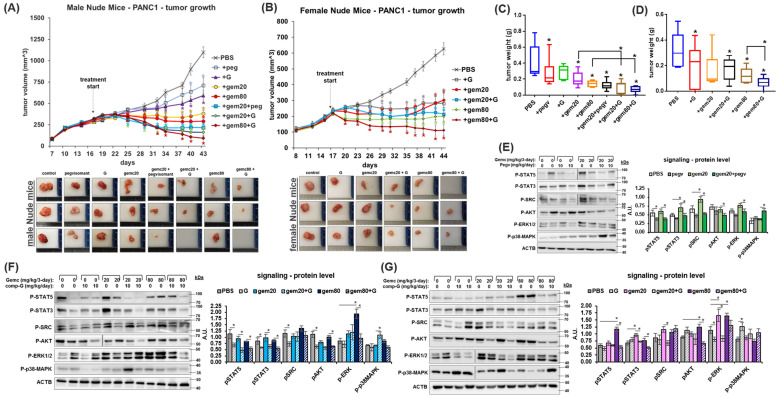
GHR antagonism improves chemotherapeutic efficacy in pancreatic cancer xenografts. (**A**) Changes in PANC1 xenograft tumor volume in male nude mice over time under treatment with either saline (+PBS), GHRA pegvisomant (+peg), GHRA compound G (+G), gemcitabine @20 mg/kg (+gem20), gemcitabine @20 mg/kg with pegvisomant (+gem20 +peg), gemcitabine @20 mg/kg with compound G (+gem20 +G), gemcitabine @80 mg/kg (+gem80), or gemcitabine @80 mg/kg with compound G (+gem80 +G). (**B**) Changes in PANC1 xenograft tumor volume in female mice over time under the same treatments as in (**A**), except that the two pegvisomant treatment groups are not present for female mice. Representative tumors at end-of-study from A and B are shown below the graphs. (**C**,**D**) End-of-study tumor weights in PANC1 xenograft nude male (**C**) or female (**D**) mice as in (**A**,**B**). (**E**–**G**) Western blots (representative blots and densitometric analysis) showing activation states of intracellular signaling mediators (STAT5, STAT3, SRC, AKT, ERK1/2, and p38-MAPK) downstream of GHR from tumor lysates in pegvisomant–gemcitabine-treated male mice (**E**), compound G–gemcitabine-treated male (**F**) and female (**G**) mice (n = 6). Significant differences were compared using repeated measures ANOVA combined with Bonferroni’s multiple comparison and * signifies *p* < 0.05.

**Figure 3 ijms-25-07438-f003:**
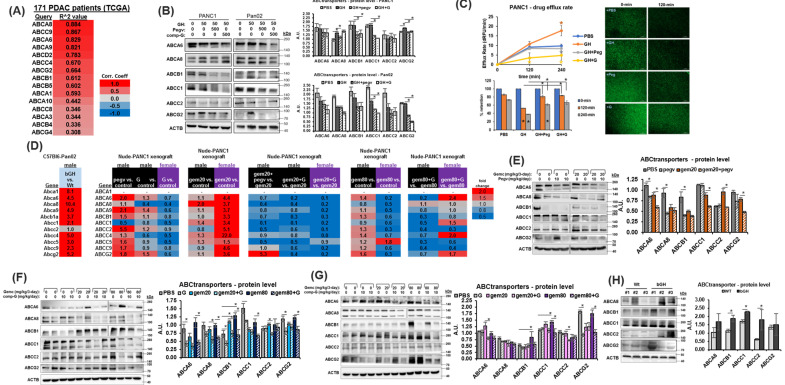
GH upregulates ABC multidrug transporter expression in PDAC: (**A**) List of top 15 (false detection rate <0.05) ABC transporter expression correlations with GHR expression in 171 human PDAC patients (TCGA database). (**B**) Western blots (representative blots and densitometric analysis) showing changes in protein levels of ABC transporters following treatment with either GH, GH + pegvisomant, or GH + compound G across all four pancreatic cancer cell lines. (**C**) (top) Cellular drug efflux rate and (bottom) cellular retention of fluorescent DiOC2 (ABC transporter substrate) in pancreatic cancer cells treated with either GH, GH + pegvisomant, or GH + compound G. (**D**) Reverse transcription and real-time quantitative PCR of RNA from xenograft tumors for relative RNA levels showing fold-changes in xenograft tumors in bGH vs. WT, and male and female nude mice with PANC1 tumors and treated with gemcitabine or GHRAs alone or in combination. (**E**–**H**) Western blots (representative blots and densitometric analysis) showing changes in tumor protein levels of ABC transporters in (**E**) PANC1 tumors in male nude mice treated with pegvisomant–gemcitabine combination, (**F**) PANC1 tumors in male nude mice treated with compound G–gemcitabine combination, (**G**) PANC1 tumors in female nude mice treated with compound-G–gemcitabine combination, or (**H**) Pan02 tumors from bGH and Wt mice. Significant differences were compared using repeated measures ANOVA combined with Bonferroni’s multiple comparison and * signifies *p* < 0.05.

**Figure 4 ijms-25-07438-f004:**
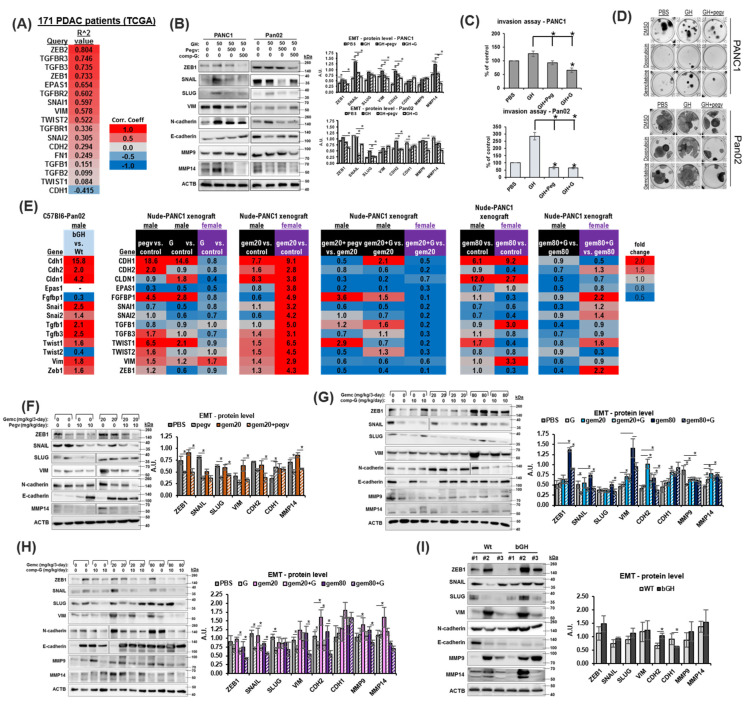
GH induces expression of epithelial-to-mesenchymal transition (EMT) factors in PDAC: (**A**) List of Pearson correlation coefficients of top 15 EMT-related genes (false detection rate < 0.05) with GHR expression in human 171 PDAC patients (TCGA database). (**B**) Western blots (representative blots and densitometric analysis) showing changes in protein levels of EMT factors following treatment with either GH, GH + pegvisomant, or GH + compound G across all four pancreatic cancer cell lines. (**C**) Effects of treatment with either GH, GH + pegvisomant, or GH + compound G on basement membrane invasion capacity of pancreatic cancer cells in culture. (**D**) Number of viable colonies formed by human and mouse pancreatic cancer cells following treatment with either GH, GH + pegvisomant, or GH + compound G in the presence/absence of chemotherapy (doxorubicin or gemcitabine). (**E**) Reverse transcription and real-time quantitative PCR of RNA from xenograft tumors for relative RNA levels and fold-changes in xenograft tumors in bGH vs. WT, and male and female nude mice with PANC1 tumors and treated with gemcitabine or GHRAs alone or in combination. (**F**–**I**) Western blots (representative blots and densitometric analysis) showing changes in tumor protein levels of EMT factors in (**F**) PANC1 tumors in male nude mice treated with pegvisomant–gemcitabine combination, (**G**) PANC1 tumors in male nude mice treated with compound G–gemcitabine combination, (**H**) PANC1 tumors in female nude mice treated with compound G–gemcitabine combination, or (**I**) Pan02 tumors from male bGH and Wt mice. Significant differences were compared using repeated measures ANOVA combined with Bonferroni’s multiple comparison and * signifies *p* < 0.05.

**Figure 5 ijms-25-07438-f005:**
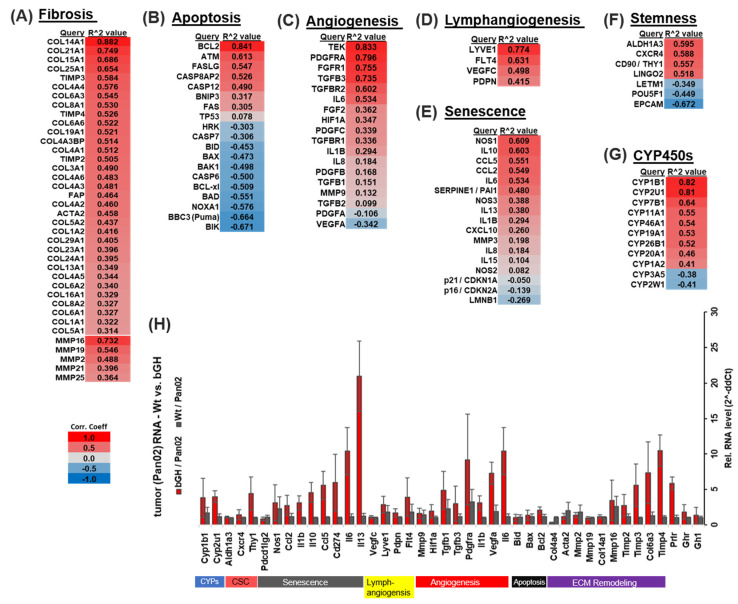
GH-induced gene expression has a multi-modal therapy refractory signature in PDAC: (**A**–**F**) List of marker genes most significantly (Pearson coefficient > 0.3, false detection rate (FDR) < 0.05) correlated with GHR expression in human PDAC patients, associated with the therapy-resistant processes (modules) of fibrosis (**A**), apoptosis (**B**), angiogenesis (**C**), lymphangiogenesis (**D**), senescence (**E**), stemness (**F**), and cytochrome P450s (**G**). (**H**) Heatmap depicting RT-qPCR cross-validation of RNA expression pattern of selected genes from each module (in (**A**–**G**)) in Pan02 tumors from WT vs. bGH male mice.

## Data Availability

All the data described in the manuscript are contained in the main text or Appendix A. Additional requests are addressed by the corresponding author. The proprietary materials used in this manuscript can be made available to other investigators with payment to cover the costs of production and reagents. All requests for protocols and proprietary material distribution should be addressed to both Nicholas Henderson (nhenderson@infinixbio.com) and John J. Kopchick (kopchick@ohio.edu).

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
