# Peer review of "Growth Hormone Receptor Antagonist Markedly Improves Gemcitabine Response in a Mouse Xenograft Model of Human Pancreatic Cancer"

_ijms, 2024, doi:10.3390/ijms25137438_

Round 1

Reviewer 1 Report

Comments and Suggestions for Authors

The article “Growth Hormone Receptor Antagonist Markedly Improves Gemcitabine Response in a Mouse Xenograft Model of Human Pancreatic Cancer” by Basu et. al, proposes a proof of concept towards considering GHR antagonist to improve chemotherapeutic outcome in the highly chemoresistant pancreatic ductal adenocarcinoma. In this study the authors discussed several aspects. By using cultured cells, mouse xenografts, and analyses of the human PDAC transcriptome, the authors identified attenuation of the multidrug transporter and epithelial-to-mesenchymal transition programs in the tumors underlie the observed augmentation of chemotherapy efficacy by GHR antagonists. It is indeed exciting that the GHR antagonists not only can reduce tumor growth but also improve treatment efficacy of anti-cancer compounds in both male and female mouse xenograft models. Thus, the authors propose that combination of GHRA with anti-cancer chemotherapy is a promising application to treat PDAC in human patients.

I find this study very interesting and addresses a broad audience in the field of cancer research. Overall the topic is relevant and the manuscript is well written, however the written style mostly addresses experts in the field. Data quality is very good, providing sufficient information in the main text as well as supplementary material. I wonder whether the authors performed certain experiments related to mutations of GH/GHR and effects of other proteins such as KRAS. Together, I have a few other concerns to be addressed(listed below) before accepting this manuscript in IJMS.

  1. Despite the detailed study presented here, the effect of mutations of the GH and GH receptor are not mentioned. Did the authors experimentally survey the effect of mutations and targeted gene deletions? It would be interesting to provide this data if experiments can be performed.

  2. Pancreatic adenocarcinoma is mostly associated with G12 residue mutation of KRAS, along with other mutations. GHR has been shown to regulate multiple cellular pathways including KRAS  signaling. Did the authors look for the correlation between KRAS and GH/GHR and the influence of gemcitabine in this perspective?

  3. References: The number of self citations are significant. Wherever applicable, I suggest to cite the original work, in addition to the review from the authors. For example lines 76-78, “We and others have recently shown that in cancers of the liver, breast, endometrium, and melanoma, GHR attenuation markedly improves chemotherapeutic efficacy in preclinical models [10].” 

  4. Lines 37-39: In the introduction authors state that Pancreatic adenocarcinoma is one of the deadliest cancers and leading cause of cancer related death in the United States. It is a major issue world wide, not only in the US. I would rather address this in a global perspective. There are studies to this end.

  5. All the figures are either of low resolution or text and axis labels are not readable. Please provide high resolution figures with larger font size. One option is to add one more row for the panel and adjust size accordingly. Relatively irrelevant figure panels can be moved to supplementary. 

  6. An extensive discussion is provided, while it is important it can be a little confusing for the readers. If allowed within the manuscript guidelines, I very much recommend separating the discussion under 2-3 sub-headings.

Author Response

We thank the reviewers for their time and assessment. We have done our best to objectively respond to each of their concerns point to point as below. Our response below follows each comment

Comment 1: Despite the detailed study presented here, the effect of mutations of the GH and GH receptor are not mentioned. Did the authors experimentally survey the effect of mutations and targeted gene deletions? It would be interesting to provide this data if experiments can be performed.

Response 1: Indeed the P495T mutation in GHR promotes lung cancer development. However, the established pancreatic cancer cell lines used in our study have publicly available gene expression data, which show that they do not have any mutations in the GH or the GHR. Therefore, we did not need to experimentally look at the effects of such mutations in our study. We also did not perform targeted gene deletions here because (i) we used peptide antagonists of GHR which is currently a more translatable method of pharmacologic inhibition of GH action, and also (ii) a 2014 study (cited in our manuscript) in cell lines of pancreatic cancer already performed targeted siRNA inhibition of GHR and observed significant suppression of cell proliferation and expression of EMT factors, which are corroborated in our study as well.

Comment 2: Pancreatic adenocarcinoma is mostly associated with G12 residue mutation of KRAS, along with other mutations. GHR has been shown to regulate multiple cellular pathways including KRAS  signaling. Did the authors look for the correlation between KRAS and GH/GHR and the influence of gemcitabine in this perspective?

Response 2: We thank the reviewers for directing us to this question as the interplay of gemcitabine in KRAS mutated cancers do have critical effects in immunotherapy response. Of the cell lines we used in our study, only PANC1 harbors the G12D-KRAS mutation (as well as a R273H-TP53 and a CDKN2A gene deletion) mutation. The other human cell line BxPC3 harbors an indel mutation in BRAF (as well as Y220C-TP53 mutation and CDKN2A and SMAD4 gene deletions). The mouse cell lines (LTPA and Pan02) do not have any RAS or RAF mutations.

Following the reviewer’s suggestion, we compared the ERK signaling under GH and GHR-antagonist effects and found that exogenous GH did not further increase endogenous ERK-phosphorylation in any of the cell lines. However, only in PANC1 cells, GHR-antagonists decreased ERK phosphorylation significantly. Therefore, we think a detailed subsequent study on GH action / inhibition across cancers with KRAS (and/or BRAF) mutations will be an exciting scope of study, which we intend to undertake in greater detail. Therefore, we thank the reviewer again for pointing it out and we intend to systematically compare this observation across cancer types and a much higher number of sample types.

Comment 3: References: The number of self-citations are significant. Wherever applicable, I suggest to cite the original work, in addition to the review from the authors. For example lines 76-78, “We and others have recently shown that in cancers of the liver, breast, endometrium, and melanoma, GHR attenuation markedly improves chemotherapeutic efficacy in preclinical models [10].” 

Response 3: We regret the number of self-citations and it is only done to limit the number of references and direct the readers to a more comprehensive coverage of the topic (as the one mentioned) wherein it includes and enlists the contributions of all members of the research community on this topic. However, we have now added original citations to the part mentioned as well as in additional relevant areas.

Comment 4: Lines 37-39: In the introduction authors state that Pancreatic adenocarcinoma is one of the deadliest cancers and leading cause of cancer related death in the United States. It is a major issue worldwide, not only in the US. I would rather address this in a global perspective. There are studies to this end.

Response 4: We agree and have now made the necessary corrections as suggested by the reviewer. The first line of Introduction has been re-written to reflect the global severity of pancreatic cancer.

Comment 5: All the figures are either of low resolution or text and axis labels are not readable. Please provide high resolution figures with larger font size. One option is to add one more row for the panel and adjust size accordingly. Relatively irrelevant figure panels can be moved to supplementary. 

Response 5: We are not sure if this was an isolated problem. We re-checked the figures, and they are >600dpi and of high resolution in the manuscript file available in the journal platform. One way would be to magnify the display scale in the pdf or word viewer to view better perhaps. A print-out of the figures will give the reader a poor resolution at this point and it would be best to view the figures on-screen than in a print-out.

Comment 6: An extensive discussion is provided, while it is important it can be a little confusing for the readers. If allowed within the manuscript guidelines, I very much recommend separating the discussion under 2-3 sub-headings.

Response 6: We have now separated the discussion under two sub-headings, as suggested by the reviewer. We agree that this improves the navigation through the discussion section and thank the reviewers for this suggestion.

Reviewer 2 Report

Comments and Suggestions for Authors

The manuscript written by Basu et al. shows the relationship between growth hormone receptor expression and tumorigenicity in pancreatic cancer, being this phenomenon reversed through two treatments with growth hormone receptor antagonists, which could generate a synergistic effect at the therapeutic level together with traditional chemotherapy (such as Gemcitabine). The results obtained in the manuscript are of high quality, very broad and with a large number of results that are able to corroborate the statements made throughout the text.

Nevertheless, there are several points that should be corrected/discussed throughout the manuscript:

1. Authors should check the in-text citations of the figures that are referenced. For example, in the in-text citation of Figures 1D and 1E (line 132), the RNA expression results indicated in the text in the cited subfigures are not shown. In addition, a western blotting experiment is performed and a table with numbers is indicated in section 1D, without being able to determine what it means.

2. Check all figure captions. They do not match the images shown in the text (e.g., in Figure 1).

3. One of the big problems in this study is the use of male and female murine models and the incorporation of different treatment patterns in both sexes, which makes many of the results not comparable. Why was this design carried out in this way and not all the experiments were carried out in the same gender? Do you think that the results obtained throughout the article can be extrapolated to both sexes? 

4. How can it be justified that the behavior is different in terms of tumor growth in the presence of growth hormone between male and female mice? Figure 1I shows that control tumors grow more than those treated with +G, while the opposite behavior is observed in females.

5. Due to the effect shown at the cytotoxicity level, what would be the treatment guideline that the authors propose for clinical trials? Do they believe that blocking this type of receptor is safe for patients, or can it give rise to other highly relevant side effects?

6. Figure 3E shows that treatment with Gemcitabine generates a decrease in the expression of most of these ABC receptors, how can this phenomenon be explained? A priori, we should think that this type of receptor is overexpressed in the presence of damage generated by a cytotoxic agent such as Gemcitabine, helping in its intracellular expulsion.

7. When reference is made to these GH inhibitor treatments, it is taken into account that all the results of overexpression of the different markers that are analyzed throughout the article are studied in tumors that express high amounts of GH. However, do you think that this therapy based on GHR inhibitors could be effective in tumor pancreatic cells with low expression? If so, would some synergy be shown?

Author Response

We thank the reviewers for their time and assessment. We have done our best to objectively respond to each of their concerns point to point as below. Our response below follows each comment:

Comment 1: Authors should check the in-text citations of the figures that are referenced. For example, in the in-text citation of Figures 1D and 1E (line 132), the RNA expression results indicated in the text in the cited subfigures are not shown. In addition, a western blotting experiment is performed and a table with numbers is indicated in section 1D, without being able to determine what it means.

Response 1: We apologize for the inadequate description in the figure legend. We corrected the figure legend. Fig 1D shows the western-blot for protein expression (top) and threshold cycle (Ct) values for the qPCR for RNA expression of the said genes in the pancreatic cancer cells. The higher the Ct value, the lower the RNA level for expression of that particular gene. The Fig-1E shows immunocytochemistry for GHR protein in the same cells.

Comment 2: Check all figure captions. They do not match the images shown in the text (e.g., in Figure 1).

Response 2: We apologize for the error. We have rechecked and corrected all figure captions, as per the reviewer’s suggestion.

Comment 3: One of the big problems in this study is the use of male and female murine models and the incorporation of different treatment patterns in both sexes, which makes many of the results not comparable. Why was this design carried out in this way and not all the experiments were carried out in the same gender? Do you think that the results obtained throughout the article can be extrapolated to both sexes? 

Response 3: We think that there might be some confusion due to inadequate figure legend. The male and the female treatment patterns are similar with the following 6 treatment groups in both:

  1. Saline / PBS (control) = PBS
  2. Compound-G (GHR antagonist) = G
  3. Gemcitabine @20mg/kg = gem20
  4. Gemcitabine @20mg/kg + compound-G = gem20+G
  5. Gemcitabine @80mg/kg = gem80
  6. Gemcitabine @80mg/kg + compound-G = gem80+G

The male mice have 2 additional treatment groups –

  1. Pegvisomant (GHR-antagonist) = peg
  2. Gemcitabine @20mg/kg + pegvisomant = gem20+peg

Therefore, the male mice were assessed with two different GHR antagonists (compound-G and pegvisomant) at the lower dose of gemcitabine. The female mice did not have the treatment groups 7 and 8 as we did not have adequate pegvisomant for both sexes. We apologize for the confusion and have now added better details in the figure legend for Fig-1 and Fig-2.

Comment 4: How can it be justified that the behavior is different in terms of tumor growth in the presence of growth hormone between male and female mice? Figure 1I shows that control tumors grow more than those treated with +G, while the opposite behavior is observed in females.

Response 4: We think that there might be some confusion due to inadequate figure legend. The letter G does not indicate growth hormone, but indicates the GHR antagonist compound-G. The result of application of GHR antagonist (compound-G) is similar in male (Fig 1I) and female (Fig 1J) mice – decrease of growth rate compared to saline (PBS) treated mice. We have added details to the figure legend to clarify better.

Comment 5: Due to the effect shown at the cytotoxicity level, what would be the treatment guideline that the authors propose for clinical trials? Do they believe that blocking this type of receptor is safe for patients, or can it give rise to other highly relevant side effects?

Response 5: We thank the reviewer’s for this important question. Our results indicate that chemotherapy efficacy (killing tumor cells) is increased when a GHR antagonist is co-administered. Therefore, the inference from this study for a future clinical trial would be co-administration of specific doses of a GHR-antagonist (e.g. FDA approved GHR-antagonist pegvisomant) with prescribed chemotherapy regimen. Currently, pegvisomant (prescribed for acromegaly) is administered in acromegaly patients at doses ranging from 10-30mg/day as a self-administered sub-cutaneous daily injection. We expect a treatment regimen of similar or lower frequency of GHR antagonist application as well in cancer patients. Several new designs and formulations of long-acting GHR-antagonists are in active research and development. Further studies are planned, and some are underway to further assess if a pre-treatment with GHR-inhibitors can decelerate malignancy or if extended treatment with GHR-inhibitors can suppress relapse and metastasis.

Comment 6: Figure 3E shows that treatment with Gemcitabine generates a decrease in the expression of most of these ABC receptors, how can this phenomenon be explained? A priori, we should think that this type of receptor is overexpressed in the presence of damage generated by a cytotoxic agent such as Gemcitabine, helping in its intracellular expulsion.

Response 6: We thank the reviewer’s for this detailed observation. Indeed as shown in Fig-3E, gemcitabine (@20mg/kg) treatment does not change and often decrease the ABC transporter levels in the tumor cells. However, as shown in Fig-3F and 3G, at a higher dose of 80mg/kg of gemcitabine, ABC transporter levels are upregulated. Therefore, this is possibly a dose-dependent effect of the chemotherapy and is observed for different chemotherapies across different cancer types.

Comment 7: When reference is made to these GH inhibitor treatments, it is considered that all the results of overexpression of the different markers that are analyzed throughout the article are studied in tumors that express high amounts of GH. However, do you think that this therapy based on GHR inhibitors could be effective in tumor pancreatic cells with low expression? If so, would some synergy be shown?

Response 7: We thank the reviewer for bringing up this excellent point. In this study, we have only highlighted the effectiveness of GHR inhibition with the GHR expression mostly in the xenografted tumor cells. However, there is a putatively strong effect of GH or of GHR inhibition on multiple components of the tumor microenvironment – for example the immune cells, the fibroblasts, the adipocytes, and the stem cells. In our immunodeficient mouse model, we did not inquire for these factors and also the aspect of GH regulation of tumor microenvironment itself qualifies for a large-scale study, which we have begun. Therefore, we think that the direct effect of GHR inhibition in improving cancer prognoses is a summation of its action on the tumor and non-tumor components of the tumor microenvironment. And that is why, even in tumors with low GHR expression, the inhibition of GHR expressed in non-tumor cells will promote anti-cancer effects. Additionally, GHR inhibition decreases IGF1 and improves insulin sensitivity systemically in vivo, which further favor cancer prognoses.

Round 2

Reviewer 1 Report

Comments and Suggestions for Authors

I appreciate the effort of the authors to incorporate my suggestions. However the modification of the text is not highlighted and that makes it difficult for the reviewer to go through the changes.

By including the suggested modifications, the authors have addressed my major concerns and suggestions. This is now a reasonably good article on Growth Hormone Receptor Antagonists and I support the publication of this manuscript in IJMS.